# Factors impacting the pre-analytical quality of blood cultures—Analysis at a tertiary medical center

Lucas Romann[1,2]*, Laura Werlen[3], Nikki Rommers[3], Anja Hermann[4], Isabelle Gisler[4], Stefano Bassetti[5], Roland Bingisser[6], Martin Siegemund[3,7], Tim Roloff[1,2,8], Maja Weisser[9], Veronika Muigg[1], Vladimira Hinic[1,8], Michael Osthoff[3,5], Fabian C. Franzeck[10], Adrian Egli[1,2,3,8]

1 Clinical Bacteriology and Mycology, University Hospital Basel, Basel, Switzerland, 2 Department Biomedicine, Applied Microbiology Research, University of Basel, Basel, Switzerland, 3 Department of Clinical Research, University of Basel, University Hospital Basel, Basel, Switzerland, 4 Chief Medical and Nursing Office, University Hospital Basel, Basel, Switzerland, 5 Internal Medicine, University Hospital Basel, Basel, Switzerland, 6 Emergency Department, University Hospital Basel, Basel, Switzerland, 7 Intensive Care Unit, University Hospital Basel, Basel, Switzerland, 8 Institute of Medical Microbiology, University of Zurich, Zurich, Switzerland, 9 Division of Infectious Diseases and Hospital Epidemiology, University Hospital Basel, Basel, Switzerland, 10 Department of Research and Analytics Services, University Hospital Basel, Basel, Switzerland

* Lucas1998R@gmail.com

## Abstract

### Background

Blood cultures (BC) are critical for the diagnosis of bloodstream infections, pathogen identification, and resistance testing. Guidelines recommend a blood volume of 8–10 mL per bottle as lower volumes result in decreased sensitivity. We aimed to evaluate factors for non-adherence to recommended volumes and assess the effects on diagnostic performance.

### Methods

From February to April 2020, we measured collected blood volumes by weighing all BC containers from inpatient samples at the University Hospital Basel. Information on BC volumes was merged with clinical and microbiological data, as well as nursing staff schedules. We analyzed factors associated with (i) BC sampling volume, (ii) reaching recommended volumes (≥8 mL), (iii) BC positivity, and (iv) time to positivity using linear and generalized linear mixed effect models.

### Results

We evaluated a total of 4'118 BC bottles collected from 686 patients. A total of 1'495 (36.3%) of all bottles contained the recommended filling volume of ≥8 mL. Using a central venous and arterial catheter for drawing blood resulted in an increase of filling volume by 0.26 mL (95% CI 0.10, 0.41) and 0.50 mL (95% CI 0.31, 0.69) compared to peripheral venipuncture, respectively. Each additional nursing staff working at the time of blood drawing

**Data Availability Statement:** Data cannot be shared publicly because of the patient data included in the statistical computation. Data are available from our statistician by request for

researchers who meet the criteria for access to confidential data.

**Funding:** The authors received no specific funding for this work.

**Competing interests:** The authors have declared that no competing interests exist.

was associated with 6% higher odds of achieving the recommended filling volume. We found no significant correlation between the filling volume and the positivity rate.

## Conclusion

Our results indicate critical pre-analytical quality markers linked to BC collection procedures to reach recommended collection volumes. No significant impact on the positivity rate was found.

## Introduction

Blood cultures (BC) are a critical cornerstone in the detection of bloodstream infections (BSI) and sepsis. Positive BCs are used to identify the pathogen causing BSI and enables to perform antimicrobial susceptibility testing and tailoring of antimicrobial treatment. The time to any antibiotic treatment is associated with mortality [1–4]. For this reason, any delay in the diagnostic work-up may impact the outcome.

Multiple studies showed that single BC bottles have a limited sensitivity of around 60% to detect the pathogen causing sepsis [5]. The amount of blood collected per BC bottle impacts the detection rate and diagnostic yield. Henning et al. showed that the odds of a positive BC increased by 13% per milliliter blood collected, but their analysis was only adjusted for a few factors [6]. Similarly, an increase in positivity by 1% and 3.3% per additional mL blood was described [7,8]. Up to six BC bottles were needed to detect about 95–99% of BSI [9]. Therefore, guidelines aim to standardize the BC collection and work-up and recommend to collect at least four BCs [10]. Multiple factors may impact the collected volume such as time of sampling (early shift vs. night shift), patient sex and age, and the type of the BC [6]. Due to the association of collected blood volumes and diagnostic yield, the sampling volume could also serve as a pre-analytical quality marker of the diagnostic procedure. The number of available nursing staff at the ward during the sampling timepoint and drawing procedures as potential factors affecting BC blood volume are so far not well explored.

In the present study, we aimed to first, analyze the impact of patient, microbiological and procedural characteristics on collected BC volumes; and second, analyze whether these factors were associated with the positivity rates of BCs and the time to positivity. We hypothesized that (i) several pre-analytic factors reduce the filling volume of BCs and (ii) that lower filling volumes result in decreased diagnostic sensitivity and prolonged time to positivity.

## Material and methods

### Study setting and population

We performed a retrospective single center observational study of all consecutively collected BCs between February 13, 2020 to the April 10, 2020 (58 days) and included aerobic and anaerobic BC bottles sent to the Clinical Bacteriology and Mycology laboratory of the University Hospital Basel (UHB). The UHB is a tertiary care hospital with a total of 773 patient beds. From 2018 to 2020, a median of 48'828 BCs per year were processed.

### Ethics approval

All methods in this study were carried out in accordance with the Declaration of Helsinki. No study specific consent, but an individual written general informed consent was available.

Patients who decline the general consent were excluded. For cases with no positive or no negative statement (unclear status), the need for consent was waived by the ethical committee as there was no ethical concern and this is a paragraph in the Swiss Human Research Act (article 34, "Humanforschungsgesetz"). Therefore, we included all BCs from hospitalized patients ≥ 18 years of age who did not reject the general informed consent. Samples from children were excluded due to different procedures and age-dependent variability in collected blood volumes.

The project was ethically evaluated and approved by the "Ethikkommission Nordwest- und Zentralschweiz (https://www.eknz.ch/)"), this is the legal body to evaluate projects in our region. The evaluation document had the following number (project-ID: 2020–00451).

## Blood culture characteristics and collection

Internal hospital standard operating procedures specify that nursing staff should fill two Monovette® tubes (Sarstedt, Nuembrecht, Germany) with 8–10 mL of blood by single sampling strategy taken under aseptic conditions without air supply and transfer the blood from each tube to an aerobic and an anaerobic BC. BC bottles are immediately sent to the laboratory for further processing.

The BC systems used are the BacT/ALERT FA FAN ® Aerobic and BacT/ALERT FN FAN® Anaerobic (bioMérieux, Roissy CDG, France) bottles. Aerobic and anaerobic bottles contain 30 mL and 40 mL of nutrient agar, respectively. BC bottles were incubated in VirtuO BacT/ALERT devices (bioMérieux) until positive or for a maximum of six days. The time to positivity was documented. Positive bottles were analyzed with subsequent standard microbiological methods [11].

Each BC was individually weighed to 0.1 g accuracy using an electronic high-precision scale (PM100, Mettler-Toledo, Greifensee, Switzerland). The blood volume (in mL) was calculated according to Henning et. al [6]. Briefly, estimated volume = (weight of bottle filled with blood [g]– mean weight of standard bottles without blood [g]) / density of blood (1.055 [g/mL]). We determined the median weight and IQR of 20 aerobic and anaerobic BCs bottles without blood, which was 61.6 g (IQR 61.5–61.6) and 72.15 g (IQR 71.7–72.3), respectively.

## Data collection

For every BC taken we collected demographic and clinical data from the respective patients' hospitalization recorded within the electronic chart via the hospital's data warehouse solution. The number of nursing staff present on wards at the timepoint of blood draw was extracted from the staff time recording system.

At the patient level, we collected age (in years), sex (male/female/unknown), and insurance type (compulsory, semi-private, or private). At the case level, we collected the type of care (in- or outpatient), infection status (yes/no), duration of stay (in days), staying at least once in the ICU during the hospital stay (yes/no), presence of at least one active infection (ICD-10) (yes/no), presence of at least one hematological or oncological active diagnosis (ICD-10) (yes/no), number of BCs sampled (frequency) and total sample volume (in mL), and the patient outcome (case closed and patient alive, patient died during case, and case not closed). At the sample level, we collected hospital ward at the time of sampling, time passed since hospital entry (in days), work shift at time of collection (early shift 7:00–14:59, late shift 15:00–22:59, and night shift 23:00–6:59), day of the week (weekday and weekend), drawing procedure (peripheral venous, central venous, arterial and unknown), total blood volume until time of sample (in mL), total blood volume per BC sample (in mL), and antibiotic administration within the

last 72 hours (yes/no). At the bottle level, we collected the bottle type (aerobic vs. anaerobic), number of nursing staff working on the ward when the BC was drawn, type of infection (none, contamination, Gram-positive bacteria, Gram-negative bacteria, Candida, and polymicrobial infection), BC filling volume (in mL), infection status (yes/no), and time to positivity (in hours).

## Definitions

Case: The duration of stay from hospital entry to discharge from hospital.

Sample: A pair of an aerobic and an anaerobic BC bottle.

Total sample volume: Includes blood from an aerobe and an anaerobe BC bottle.

Total case volume: Includes the blood of a patient collected for BCs during the whole length of a hospital stay.

Total blood volume until time of sample: Collected blood volume during the case until the BC we were looking at was drawn. Including the present sample.

Infection status: A BC is positive or there is a positive BC during the case (contaminated BCs were excluded).

At least one infectious diagnosis: There has been at least one infection during the whole hospital stay of a patient according to the ICD 10 diagnosis with which the patient was labeled. It does not differ regarding the type of infection.

## Contamination

**Definition.**  Blood cultures were considered to be contaminated based on criteria described by Elzi et al. [12]. Briefly, the following criteria were used: (i) if two or more blood cultures were positive within a 7-day period, it is probably a true BSI; (ii) if the patient has a central venous catheter and only one positive BC, two or more SIRS criteria (as defined [13]) are needed for a true BSI; and (iii) if the patient has no central venous catheter and only one positive BC, three or more SIRS criteria are needed for a true BSI. If these criteria (i,ii or iii) were not fulfilled the BC was contaminated. Positive BC triggered a written evaluation by the infectious disease team. The classification regarding contamination of a BC bottle was adopted from this assessment.

**Contamination rate.**  Our contamination rate was 1.3% of all BC that were taken. Within the positive BC about 22.4% were contaminated. According to a different study [14], a benchmark of 1.5%-2% contaminated BC is standard and also attainable. Our data is limited due to the fact that we had to filter the contaminated bottles by hand. For that reason, we took only the common contamination microorganisms within our search. If one of these microorganisms were detected the probe was seen by an infectiologist who determined if the BC was contaminated or not. The common contamination microorganisms for which we corrected manually were: *Staphylococcus aureus*, *Escherichia coli*, *Enterococcus faecium*, *Klebsiella pneumoniae*, *Enterococcus faecialis*, *Pseudomonas spp.*, ESBL, *Streptococcus pneumoniae*, *Citrobacter freundii-group*, *Morganella morganii*, *Enterobacter cloacae-group*, and *Campylobacter jejuni*.

## Study endpoints

The primary endpoints were: (i) the filling volume of BC bottles in mL and (ii) BCs fulfilling the recommended volumes of at least 8 mL. Secondary endpoints were (i) positivity of BCs and (ii) the time to positivity of positive BCs.

## Statistical analyses

All analyses were performed in R version 4.0.3 (2020-10-10) [15]. No prior power calculation was carried out beforehand. For the analyses we used mixed-effects models. Specifically, we used linear mixed-effects models for the primary outcome bottle volume and secondary outcome natural logarithm of time to positivity. We used mixed-effects logistic regression models for the outcomes recommended filling volume and positivity. The independent variables pre-specified in the analytical plan specific to each model were modeled as fixed effects. We accounted for the repetitive sampling of individual patients at several time points by including pseudonymized patient and case ID as random effects, with case nested within patient. For the analysis of the secondary endpoints, we could not include all variables specified due to non-convergence of the model. The excluded variables were: patient sex, type of care, hospital ward, number of nursing staff, insurance type, bottle type, and work shift.

To build and analyze the mixed-effects models, we used the R package lme4 [16]. We calculated the confidence intervals for the linear mixed effects models using the basic bootstrap method with 1'000 simulations. Due to convergence issues of the logistic regression models, we used a non-parametric bootstrapping method with 1'000 simulations that resamples based on the different levels of the random effects to estimate the confidence intervals.

The results shown are from our multivariable model, in which all covariates listed were included, and the coefficients reported the estimated average effects after adjusting for all other variables in the model. Results of the univariate model are presented in the tables.

## Results

From January 2019 to December 2020 a median of 4'122 BCs (IQR 3'971–4'357) were processed monthly at the UHB with an average positivity rate of 8.4%. During the study period, we collected a total of 4'456 BCs from 791 patient cases and excluded 65 patients who declined the general informed consent for research. On the date of data extraction (September 9, 2020), only 35 cases had not yet been finalized. For the final analysis, we included a total of 4'118 BCs from 726 cases of 686 patients (S1 Fig). Out of the 4'118 BCs collected, 245 BCs (5.9%) were positive and 3'873 BCs (94.1%) were negative. Positive BCs did not show significantly higher volumes compared to negative BCs (7.34 mL, IQR 7.15–7.55 vs. 7.17 mL, IQR 7.13–7.22).

### Baseline characteristics

Table 1 summarizes the detailed characteristics of patients, cases, samples, and BCs (Table 1). The median patient age in our cohort was 67 years (IQR 51–78) and 291/686 (42.4%) were female.

**Cases.** Out of a total of 726 cases, 40 (5.5%) had a BSI. The median duration of hospital stay for a case was 7.7 days (IQR 3.0–16.3). 192/726 (26.4%) cases included at least one ICU stay with a median length of ICU stay of 3.5 days (IQR 1.5–8.3). In-hospital death occurred in 47/726 (6.5%) of cases.

**Samples.** We collected data on a total of 2'099 samples, usually an aerobic/anaerobic pair of BC bottles. Overall, 590/2'099 (28.1%) samples were drawn from patients hospitalized in medicine wards, 767/2'099 (36.5%) on the emergency department, and 396/2'099 (18.9%) on the intensive care unit. The most frequent drawing procedure was from a peripheral vein (895/2'099, 42.6%). The median collected blood volume per sample was 14.9 mL (IQR 12.5–16.4). In 811/2'099 (38.6%) of drawn samples, there has been an antibiotic administration within the last 72 hours prior to the blood collection and 182 (8.7%) samples had no antibiotics. However, we did not have access to detailed information on antibiotic usage in the 72 hours before the

**Table 1. Basic characteristics for each patient, case, sample and BC bottles.**

| Patient | Overall | Missing (%) |
|---|---|---|
| n | 686 | |
| Patient age (median [IQR]) | 67.0 [51.0, 78.0] | 0 |
| Patient sex = female (%) | 291 (42.4) | 0 |
| Insurance type (%) | | 0 |
| • compulsory | 576 (84.0) | |
| • semi-private | 70 (10.2) | |
| • private | 40 (5.8) | |
| **Case** | | |
| n | 726 | |
| Type of care = outpatient (%) | 108 (14.9) | 0 |
| Positivity = Infection (%) | 40 (5.5) | 0 |
| Duration of stay (in days) (median [IQR]) | 7.7 [3, 16.3] | 4.8 |
| ICU stay = Yes (%) | 192 (26.4) | 0 |
| Days spent in ICU among those who were in ICU (median [IQR]) | 3.5 [1.5, 8.3] | 75.5 |
| At least one infectious diagnosis = Yes (%) | 521 (71.8) | 0 |
| At least one haematological or oncological diagnosis = Yes (%) | 274 (37.7) | 0 |
| Bottles sampled in total (median [IQR]) | 4.0 [4.0, 8.0] | 0 |
| Total sample volume over case (in mL) (median [IQR]) | 31.2 [24.9, 46.4] | 0 |
| Patient outcome (%) | | 0 |
| • case closed, patient lived | 644 (88.7) | |
| • patient died during case | 47 (6.5) | |
| • case not closed | 35 (4.8) | |
| **Sample** | | |
| n | 2099 | |
| Hospital ward (%) | | 0 |
| • medicine | 590 (28.1) | |
| • emergency | 767 (36.5) | |
| • intensive care | 396 (18.9) | |
| • speciality clinic | 118 (5.6) | |
| • surgery | 201 (9.6) | |
| • unknown | 27 (1.3) | |
| Time passed since hospital entry (in days) (median [IQR]) | 1.0 [0.0, 7.1] | 0 |
| Work shift (%) | | 0 |
| • early | 911 (43.4) | |
| • late | 938 (44.7) | |
| • night | 250 (11.9) | |
| Weekend vs. weekday = weekend (%) | 481 (22.9) | 0 |
| Drawing procedure (%) | | 0 |
| • peripheral venous | 895 (42.6) | |
| • central venous | 321 (15.3) | |
| • arterial | 141 (6.7) | |
| • unknown | 742 (35.4) | |
| Total blood volume per sample (median [IQR]) | 14.9 [12.5, 16.4] | 0 |
| Total blood volume until time of sample (median [IQR]) | 31.8 [24.5, 50.3] | 0 |
| Antibiotic administration within last 72 hours (%) | | 0 |
| • No | 182 (8.7) | |
| • Yes | 811 (38.6) | |
| • Not assessable | 1106 (52.7) | |

(*Continued*)

**Table 1.** (Continued)

| Patient | Overall | Missing (%) |
|---|---|---|
| **Bottle** | | |
| n | 4118 | |
| Bottle type = anaerobic (%) | 2055 (49.9) | 0 |
| Number of staff working on station (median [IQR]) | 11.00 [7.00, 16.00] | 3.8 |
| Type of infection (%) | | 0 |
| • No infection identified | 3873 (94.1) | |
| • Contamination | 55 (1.3) | |
| • Gram positive [a] | 104 (2.5) | |
| • Gram negative [a] | 71 (1.7) | |
| • Candida spp. [a] | 5 (0.1) | |
| • Polymicrobial infection | 10 (0.2) | |
| Bottle volume (in mL) (median [IQR]) | 7.5 [6.4, 8.3] | 0 |
| Recommended volume (in mL) = $\geq$ 8 mL (%) | 1495 (36.3) | 0 |
| Positivity = Infection (%) | 190 (4.6) | 0 |
| Time to positivity (in hours) (median [IQR]) | 14.8 [11.1, 22.4] | 0.5 |
| Typical time to positivity (categorical) (%) | | 0.3 |
| < 20 hours | 149 (3.6) | |
| 20–30 hours | 71 (1.7) | |
| > 30 hours | 6 (0.1) | |
| No infection | 3878 (94.2) | |
| NA | 14 (0.3) | |

[a] For further details see S1 Table.

BCs were taken in the majority of the samples (52.7%), as this period commonly began before the hospitalization.

**Bottles.** In the majority of included bottles—3'873/4'118 (94.1%)—no growth of microorganisms was observed. Bacterial growth was identified in 245/4'118 (5.9%) bottles, of which 55/245 (22.4%) were judged to represent a contaminated BC. We documented a median filling volume of 7.5 mL (IQR 6.4–8.3) per bottle. Only 1'495/4'118 (36.3%) bottles were filled to the recommended volume of 8 mL or more (S2 Fig). Only 14 BC bottles had a filling volume above 10 mL. The median time to positivity was 14.8 hours (IQR 11.1–22.4).

## Factors associated with BC filling volume

First, we analyzed the association of factors with BC filling volume (primary endpoint) and used a uni- and multivariable model (Fig 1, Table 2). Results from the uni- and multivariable models were similar. The central venous and arterial drawing procedure, in comparison to the peripheral venous drawing procedure, had on average a 0.26 mL (95% CI 0.10, 0.41) and 0.50 mL (95% CI 0.31, 0.69) higher filling volume, respectively. Anaerobic bottles were associated with an average 0.16 mL (95% CI -0.23, -0.08) lower filling volume compared to aerobic bottles. Each additional staff person working at the time of blood drawing was associated with an additional 0.03 mL of collected blood (95% CI 0.01, 0.04).

## Factors associated with reaching the recommended filling volume

Next, we analyzed the factors associated with the recommended bottle volume of 8 mL or more (primary endpoint) using a uni- and multivariable model (Fig 2, Table 3). Results from

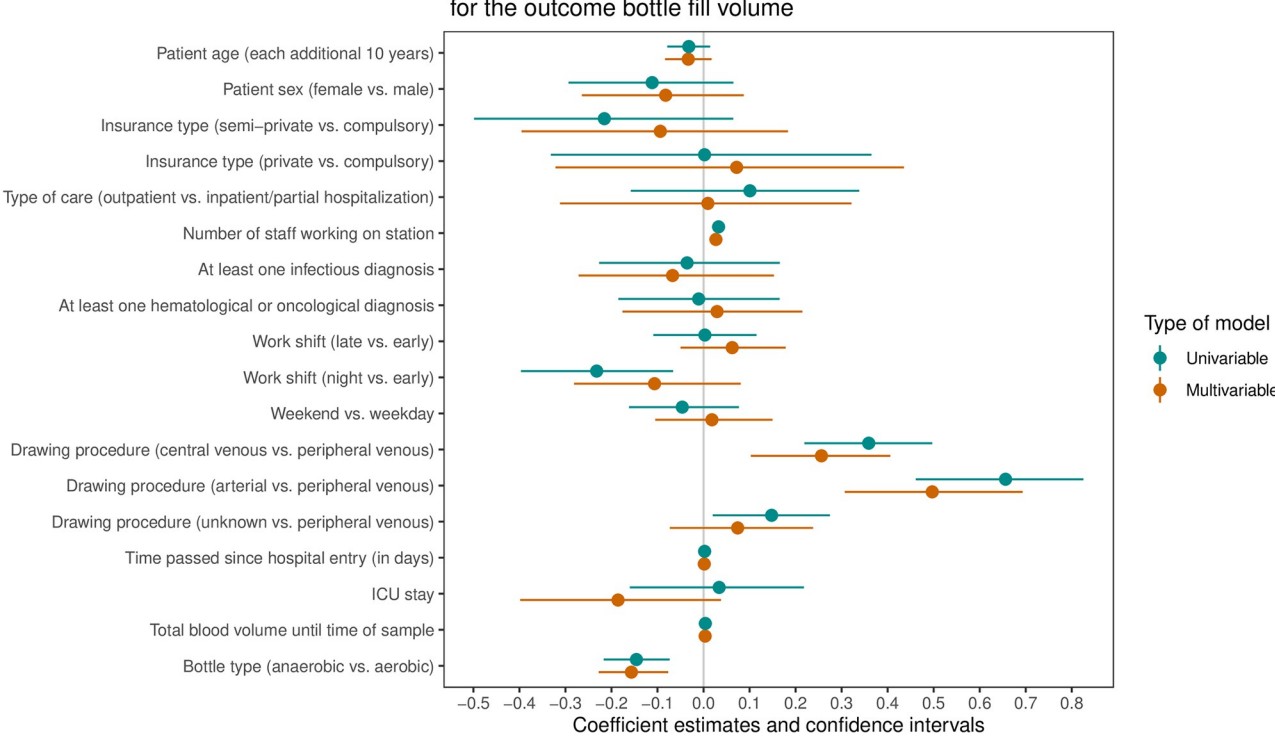

**Fig 1. Association of selected exposure variables with blood culture filling volume (in mL) in uni- and multivariable mixed linear regression models.**

the uni- and multivariable models are similar. Blood samples drawn in women showed a lower odds ratio (OR) of 0.71 (95% CI 0.52, 0.95) for reaching the recommended filling volume compared to blood samples drawn in men.

The arterial drawing procedure was associated with higher odds of achieving the recommended filling volume as compared to the peripheral venous drawing procedure with an odds ratio of 2.08 (95% CI 1.31, 2.87), respectively. Each additional staff working at the time of blood drawing was associated with an odds ratio of 1.06 (95% CI 1.04, 1.09) of achieving the recommended filling volume. BCs taken from patients with at least one ICU stay during a case were associated with lower odds of achieving the recommended filling volume than patients without an ICU stay (odds ratio of 0.73, 95% CI 0.49, 0.97) when adjusted for all other variables in the model. The odds ratio of having the recommended blood volume was 0.52 (95% CI 0.44, 0.61) for anaerobic bottles compared to aerobic bottles.

## Factors associated with positivity of BC

In the multivariable model, none of the covariates including the BC filling volume remained associated with positivity (Table 4, S3 and S4 Figs).

## Factors associated with time to positivity

No significant associations with time to positivity were observed. This applies also for the BC filling volume (Table 5, S5 Fig).

**Table 2. Uni- and multivariable results of associated factors with "bottle filling volume".**

| | Univariable model | | Multivariable model | |
|---|---|---|---|---|
| | Coefficient | 95% CI | Coefficient | 95% CI |
| **Patient** | | | | |
| Patient age | 0.00 | [-0.01, 0.00] | 0.00 | [-0.01, 0.00] |
| Patient sex (female vs. male) | -0.11 | [-0.29, 0.06] | -0.08 | [-0.26, 0.09] |
| Insurance type (semi-private vs. compulsory) | -0.22 | [-0.50, 0.06] | -0.09 | [-0.40, 0.18] |
| Insurance type (private vs. compulsory) | 0.00 | [-0.33, 0.36] | 0.07 | [-0.32, 0.44] |
| **Case** | | | | |
| Type of care (outpatient vs. inpatient/partial hospitalization) | 0.10 | [-0.16, 0.34] | 0.01 | [-0.31, 0.32] |
| ICU stay | 0.03 | [-0.16, 0.22] | -0.19 | [-0.40, 0.04] |
| At least one infectious diagnosis | -0.04 | [-0.23, 0.17] | -0.07 | [-0.27, 0.15] |
| At least one haematological or oncological diagnosis | -0.01 | [-0.19, 0.17] | 0.03 | [-0.18, 0.21] |
| **Sample** | | | | |
| Time passed since hospital entry (in days) | 0.00 | [-0.00, 0.00] | 0.00 | [-0.00, 0.00] |
| Work shift (late vs. early) | 0.00 | [-0.11, 0.12] | 0.06 | [-0.05, 0.18] |
| Work shift (night vs. early) | -0.23 | [-0.40, -0.07] | -0.11 | [-0.28, 0.08] |
| Weekend vs. weekday | -0.05 | [-0.16, 0.08] | 0.02 | [-0.10, 0.15] |
| Drawing procedure (central venous vs. peripheral venous) | 0.36 | [0.22, 0.50] | 0.26 | [0.10, 0.41] |
| Drawing procedure (arterial vs. peripheral venous) | 0.66 | [0.46, 0.83] | 0.50 | [0.31, 0.69] |
| Drawing procedure (unknown vs. peripheral venous) | 0.15 | [0.02, 0.27] | 0.07 | [-0.07, 0.24] |
| Total blood volume until time of sample | 0.00 | [0.00, 0.01] | 0.00 | [0.00, 0.00] |
| **Bottle** | | | | |
| Bottle type (anaerobic vs. aerobic) | -0.15 | [-0.22, -0.07] | -0.16 | [-0.23, -0.08] |
| Number of staff working on station | 0.03 | [0.02, 0.04] | 0.03 | [0.01, 0.04] |

## Discussion

In this study, we investigated (i) possible factors that may have an impact on the filling volume of BCs and (ii) the relationship between patient, microbiological and procedural characteristics on positivity rate and time to positivity. Our key findings show that the majority of BCs were filled below the recommended volume. BCs from female patients have a lower likelihood of reaching the recommended filling volume, while additional nursing staff are associated with higher odds. Furthermore, drawing procedures have an impact on the filling volume. Filling volume was not associated with BC positivity in the multivariable model.

Only 36.3% of collected BC bottles fulfilled the recommended BC filling volumes of 8 to 10 mL per bottle. Similar results have been shown in previous studies [6,17–19].

We observed a significantly lower rate of reaching the recommended volume filling of at least 8mL per mL in female patients, but in contrast to Henning et al., found no effect of age [6]. The reason for these differences relating to patient sex is unknown to us.

A recent survey by Elvy et al. summarized the clinical practice of BC collection in Australia and New Zealand. The authors investigated pre-analytical differences across centers and noted that nurses collected the samples in 32% of the cases, trained phlebotomists in 30%, doctors in 17% and not stated in 21% [20]. In our center, blood for BCs were drawn by trained nurses only. To our knowledge, this is the first study to examine the effects of available nursing staff working at a ward and its impact on the recommended filling volume. Every additional staff person working during the time of the BC collection increased the odds ratio of achieving the recommended filling volume by 6%. The amount of nursing staff available may impact the pre-analytical quality of BC collection. While the effect is small for adding just one additional

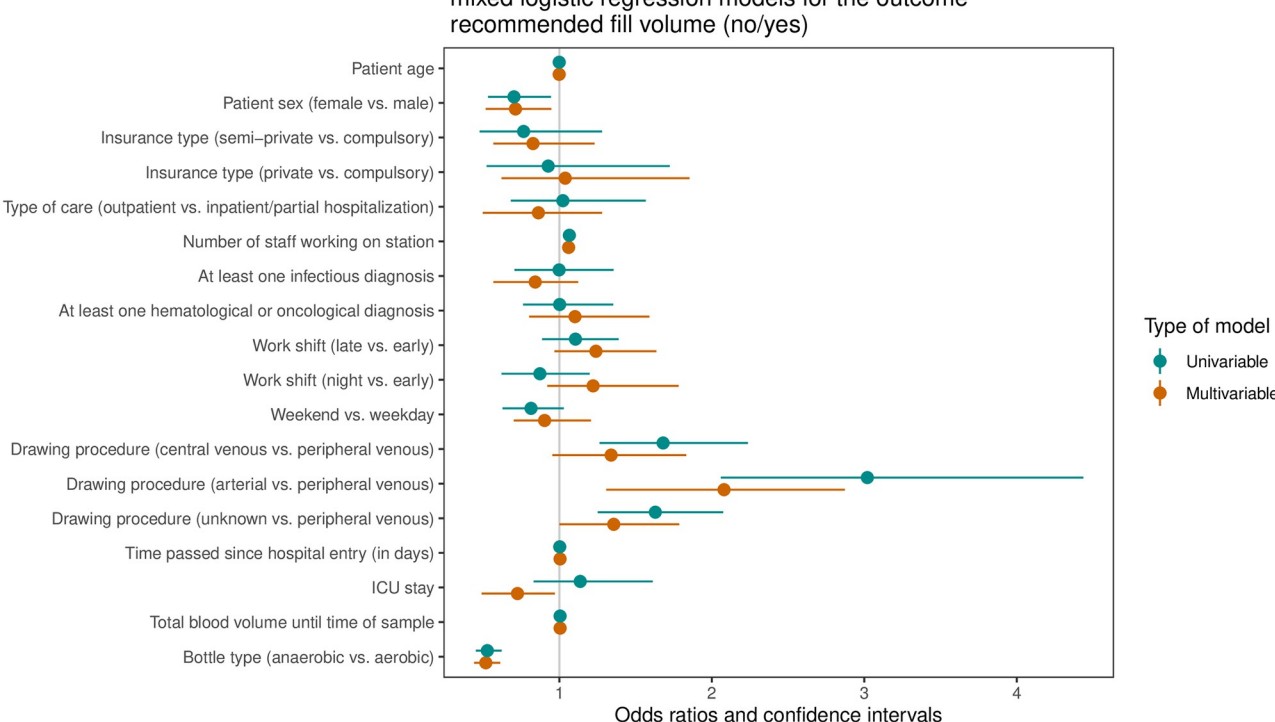

**Fig 2. Association of prespecified exposure variables with achieving the recommended blood volume per bottle of $\geq$ 8 mL in uni- and multivariable mixed logistic regression models.**

nursing staff per single BC, adding several staff members may obviously show a potentially larger effect. This effect may be especially important during night shifts. Henning et al. reported that during night shift in average 8.2 mL were collected and significantly more during daytimes with 8.68 mL [6]. Nonetheless, we were not able to demonstrate that BCs collected during the night shift had lower filling volumes than those collected during the early shift. Various reasons are conceivable, for example different handling of taking a BC or the lack of enough data to show a significant difference. Therefore, increasing the number of nurses in clinical practice would not be justifiable, because the effect is limited and the expenses too costly. Moreover, it is not the only reason for the lower filling volumes.

We were able to show that the drawing procedure had a substantial impact on the filling volume of BCs. The peripheral venous drawing procedure was associated with significantly lower filling volumes than the central venous or the arterial drawing procedures. To our knowledge, this has not yet been studied and may be for the reason that blood collection via a central venous or arterial catheter is easier because the procedure is performed on an already existing catheter rather than on the freshly punctured vein. Yet, they are more costly and might require a specialist placing them, which reflects just the real-world experience. Nonetheless, contaminations of intravenous catheters have been successfully reduced in the last years with different methods [21,22]. To evaluate the cost-effectiveness further studies are needed.

The rate of positivity was not influenced by any factors. Of interest was the absence of a significant association of BC filling volume with positivity in our multivariable analysis. Other studies reported significant effects, however comparison must be made with caution, as patient selection and specification of multivariable regression models differed. Furthermore, reported

**Table 3. Uni- and multivariable results of associated factors with "recommended bottle filling volume".**

| | Univariable model | | Multivariable model | |
| --- | --- | --- | --- | --- |
| | **OR** | **95% CI** | **OR** | **95% CI** |
| **Patient** | | | | |
| Patient age | 1.00 | [0.99, 1.01] | 1.00 | [0.99, 1.01] |
| Patient sex (female vs. male) | 0.70 | [0.53, 0.95] | 0.71 | [0.52, 0.95] |
| Insurance type (semi-private vs. compulsory) | 0.77 | [0.48, 1.28] | 0.83 | [0.57, 1.23] |
| Insurance type (private vs. compulsory) | 0.93 | [0.52, 1.72] | 1.04 | [0.62, 1.85] |
| **Case** | | | | |
| Type of care (outpatient vs. inpatient/partial hospitalization) | 1.02 | [0.68, 1.57] | 0.86 | [0.50, 1.28] |
| ICU stay | 1.14 | [0.83, 1.61] | 0.73 | [0.49, 0.97] |
| At least one infectious diagnosis | 1.00 | [0.71, 1.36] | 0.84 | [0.57, 1.12] |
| At least one haematological or oncological diagnosis | 1.00 | [0.76, 1.35] | 1.10 | [0.80, 1.59] |
| **Sample** | | | | |
| Time passed since hospital entry (in days) | 1.00 | [1.00, 1.01] | 1.00 | [1.00, 1.01] |
| Work shift (late vs. early) | 1.11 | [0.89, 1.39] | 1.24 | [0.97, 1.64] |
| Work shift (night vs. early) | 0.87 | [0.62, 1.20] | 1.22 | [0.92, 1.78] |
| Weekend vs. weekday | 0.81 | [0.63, 1.03] | 0.90 | [0.70, 1.21] |
| Drawing procedure (central venous vs. peripheral venous) | 1.68 | [1.26, 2.24] | 1.34 | [0.95, 1.83] |
| Drawing procedure (arterial vs. peripheral venous) | 3.02 | [2.06, 4.44] | 2.08 | [1.31, 2.87] |
| Drawing procedure (unknown vs. peripheral venous) | 1.63 | [1.25, 2.07] | 1.36 | [1.00, 1.79] |
| Total blood volume until time of sample | 1.01 | [1.00, 1.01] | 1.01 | [1.00, 1.01] |
| **Bottle** | | | | |
| Bottle type (anaerobic vs. aerobic) | 0.53 | [0.45, 0.62] | 0.52 | [0.44, 0.61] |
| Number of staff working on station | 1.07 | [1.05, 1.08] | 1.06 | [1.04, 1.09] |

**Table 4. Uni- and multivariable results of associated factors with "positivity".**

| | Univariable model | | Multivariable model | |
| --- | --- | --- | --- | --- |
| | **OR** | **95% CI** | **OR** | **95% CI** |
| **Patient** | | | | |
| Patient age | 1.01 | [1.00, 1.03] | 1.01 | [0.89, 1.19] |
| **Case** | | | | |
| ICU stay | 1.36 | [0.41, 8.23] | 2.36 | [0.16, 1024.18] |
| At least one haematological or oncological diagnosis | 1.57 | [1.02, 5.60] | 2.77 | [0.96, 1579835779401.85] |
| **Sample** | | | | |
| Time passed since hospital entry (in days) | 1.01 | [0.91, 1.07] | 1.01 | [0.98, 1.54] |
| Drawing procedure (central venous vs. peripheral venous) | 0.68 | [0.11, 3.24] | 0.56 | [0.17, 3.90] |
| Drawing procedure (arterial vs. peripheral venous) | 0.72 | [0.08, 5.68] | 0.74 | [0.00, 7.72] |
| Drawing procedure (unknown vs. peripheral venous) | 2.34 | [0.34, 45.04] | 0.64 | [0.00, 63.87] |
| Total blood volume until time of sample | 1.00 | [0.96, 1.01] | 1.00 | [0.95, 1.02] |
| Antibiotic administration within last 72 hours (yes vs. no) | 0.17 | [0.01, 1.31] | 0.13 | [0.01, 1.86] |
| Antibiotic administration within last 72 hours (not assessable vs. no) | 3.33 | [0.44, 238.47] | 1.85 | [0.57, 10541.97] |
| **Bottle** | | | | |
| Blood volume (in mL) | 1.14 | [0.84, 1.50] | 1.06 | [0.79, 1.53] |

**Table 5. Uni- and multivariable results of associated factors with "time to positivity".**

| | Univariable model | | Multivariable model | |
|---|---|---|---|---|
| | Estimate | 95% CI | Estimate | 95% CI |
| **Patient** | | | | |
| Patient age | 1.00 | [0.99, 1.00] | 1.00 | [1.00, 1.01] |
| **Case** | | | | |
| ICU stay | 1.03 | [0.81, 1.27] | 0.83 | [0.69, 1.02] |
| At least one infectious diagnosis | 0.95 | [0.63, 1.41] | 0.89 | [0.65, 1.21] |
| At least one haematological or oncological diagnosis | 0.87 | [0.69, 1.09] | 0.84 | [0.68, 1.05] |
| **Sample** | | | | |
| Time passed since hospital entry (in days) | 1.00 | [0.99, 1.01] | 0.99 | [0.98, 1.00] |
| Drawing procedure (central venous vs. peripheral venous) | 1.14 | [0.97, 1.35] | 0.90 | [0.78, 1.05] |
| Drawing procedure (arterial vs. peripheral venous) | 0.82 | [0.56, 1.21] | 0.95 | [0.69, 1.34] |
| Drawing procedure (unknown vs. peripheral venous) | 0.82 | [0.69, 0.99] | 0.94 | [0.78, 1.12] |
| Total blood volume until time of sample | 1.00 | [1.00, 1.00] | 1.00 | [1.00, 1.00] |
| Antibiotic administration within last 72 hours (yes vs. no) | 0.90 | [0.71, 1.15] | 1.16 | [0.90, 1.49] |
| Antibiotic administration within last 72 hours (not assessable vs. no) | 0.61 | [0.45, 0.82] | 0.75 | [0.55, 1.03] |
| **Bottle** | | | | |
| Bottle volume (in ml) | 0.99 | [0.96, 1.03] | 1.00 | [0.97, 1.03] |
| Typical time to positivity (20–30 hours vs. < 20 hours) | 1.50 | [1.29, 1.71] | 1.42 | [1.22, 1.64] |
| Typical time to positivity (> 30 hours vs. < 20 hours) | 3.37 | [2.02, 5.60] | 3.33 | [1.84, 6.10] |

effect sizes were rather small. A recent study by Henning et al. [6] reported an OR of 1.13 (P < 0.001) per mL of increased filling volume in a multilevel regression model with the outcome of overall patient positivity (i.e., the patient had ≥1 positive BC in the dataset). They described a secondary analysis of a logistic regression with the outcome of individual bottle positivity (not accounting for within-subject clustering anymore), which resulted in an OR of 1.02 (p < 0.001) per mL.

Neves et al. [7] reported an OR 1.01 (95% CI 1.01, 1.02) per mL increase of filling volume in a multivariable model adjusting for age, gender, number of comorbidities, admission diagnosis and temperature >39 ˚C. Another study by Bouza et al. [8] did not find an independent association of filling volume and positivity in their general study population. However, they reported an OR of 1.03 (95% CI, 1.002 to 1.07, p = 0.04) for the subset of patients, who had at least one positive BC flask during their whole observation period.

Therefore, we conclude that 1) other authors also described small effects of filling volume to positivity, and 2) the precision of our estimate was most likely not able to detect such small effect sizes due to the sample size in our study.

Our study has some important limitations. First of all, the unusual distribution of blood volume per bottle is probably the result of the limited capacity of the syringes used to transfer the blood into the BC bottles, which only have a capacity of 9 mL. Therefore, they are most likely part of the reason for the low filling volume in more than half of the BC bottles. Furthermore, even if we included over 4'000 BCs, only 246 bottles were positive, which limited statistical power. We did not have enough data to analyze the effects of specific microorganisms individually. The lower positivity rate in the time of BC weighing is due to correction of multiple sampling from patients and exclusion of patients who declined the general consent. Two important variables, the drawing procedure and the antibiotic administration within the last 72 hours, had various missing data, which may be considered in a prospective study design. We did not analyze the contamination rate at the different drawing sites. However, this has been examined in

several studies [23,24], which found no significant difference between the rates of contamination among the various sites of blood draw. We did not integrate the training level and experience of the nursing staff into the model—this may be interesting to include in a future study including more BCs. The time from blood sampling to the start of the incubation of the BC was not distinguishable in our dataset. However, in our opinion the impact it had on our time to positivity was small because these times are rather short as clinical practice shows us at the UHB.

In conclusion, our study shows that there is a significant proportion of BCs which are inadequately filled according to well established guidelines. However, the actual changes in volume are rather small and there is no single clear point where a substantial improvement can be reached. The proper collection of BCs may be increased by introducing a bundle of improvement steps such as educating the employees, adding more nursing staff on a ward, or changing the type of syringes to allow larger blood volume collections. Our study found availability of nursing staff and collection procedure to be associated with the volume of blood collected.

## Supporting information

**S1 Fig. Flowchart of the BCs included into this study.**
(TIF)

**S2 Fig. Frequency distribution of the collected blood volumes per bottle.**
(PDF)

**S3 Fig. Probability of "positivity" by total sample volume (in mL) using the univariable model.**
(PDF)

**S4 Fig. Association of "positivity" and the influencing factors.**
(PDF)

**S5 Fig. Association of "time to positivity" and the influencing factors.**
(PDF)

**S1 Table. Microorganisms.**
(XLSX)

## Author Contributions

**Conceptualization:** Lucas Romann, Laura Werlen, Nikki Rommers, Anja Hermann, Isabelle Gisler, Maja Weisser, Michael Osthoff, Fabian C. Franzeck, Adrian Egli.

**Data curation:** Lucas Romann, Laura Werlen, Nikki Rommers, Anja Hermann, Isabelle Gisler, Stefano Bassetti, Roland Bingisser, Martin Siegemund, Tim Roloff, Maja Weisser, Veronika Muigg, Vladimira Hinic, Michael Osthoff, Fabian C. Franzeck, Adrian Egli.

**Formal analysis:** Laura Werlen, Nikki Rommers, Tim Roloff, Fabian C. Franzeck.

**Investigation:** Lucas Romann.

**Methodology:** Lucas Romann, Laura Werlen, Nikki Rommers, Fabian C. Franzeck, Adrian Egli.

**Project administration:** Adrian Egli.

**Resources:** Lucas Romann, Anja Hermann, Isabelle Gisler, Stefano Bassetti, Roland Bingisser, Martin Siegemund, Veronika Muigg, Michael Osthoff, Fabian C. Franzeck.

**Supervision:** Adrian Egli.

**Validation:** Adrian Egli.

**Writing – original draft:** Lucas Romann.

**Writing – review & editing:** Lucas Romann, Laura Werlen, Nikki Rommers, Stefano Bassetti, Roland Bingisser, Martin Siegemund, Tim Roloff, Maja Weisser, Veronika Muigg, Vladimira Hinic, Michael Osthoff, Fabian C. Franzeck, Adrian Egli.

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
