## [Decision Letter · Decision Letter 0]

5 Feb 2023

PONE-D-23-00409Factors impacting the pre-analytical quality of blood cultures - analysis at a tertiary medical center.PLOS ONE

Dear Dr. Lucas Romann,

Thank you for submitting your manuscript to PLOS ONE. After careful consideration, we feel that it has merit but does not fully meet PLOS ONE’s publication criteria as it currently stands. Therefore, we invite you to submit a revised version of the manuscript that addresses the points raised during the review process As you can see below your manuscript has been deeply evaluated by three expert reviwers in the filed and all of them indicate  that your paper should be revised in order to improve the overall quality of the manuscript, and I totally agree with htese suggestions. So I invite you to take the points raised into consideration and to submit a revied version of tyhis paper

Please submit your revised manuscript by two weeks:  If you will need more time than this to complete your revisions, please reply to this message or contact the journal office at plosone@plos.org. Please include the following items when submitting your revised manuscript:A rebuttal letter that responds to each point raised by the academic editor and reviewer(s). You should upload this letter as a separate file labeled 'Response to Reviewers'.A marked-up copy of your manuscript that highlights changes made to the original version. You should upload this as a separate file labeled 'Revised Manuscript with Track Changes'.An unmarked version of your revised paper without tracked changes. You should upload this as a separate file labeled 'Manuscript'.If applicable, we recommend that you deposit your laboratory protocols in protocols.io to enhance the reproducibility of your results. Protocols.io assigns your protocol its own identifier (DOI) so that it can be cited independently in the future. For instructions see: https://journals.plos.org/plosone/s/submission-guidelines#loc-laboratory-protocols. Additionally, PLOS ONE offers an option for publishing peer-reviewed Lab Protocol articles, which describe protocols hosted on protocols.io. Read more information on sharing protocols at https://plos.org/protocols?utm_medium=editorial-email&utm_source=authorletters&utm_campaign=protocols.

We look forward to receiving your revised manuscript.

Kind regards,

Vittorio Sambri, M.D., Ph.D.

Academic Editor

PLOS ONE

Journal Requirements:

Reviewers' comments:

Reviewer's Responses to Questions

**Comments to the Author**

1. Is the manuscript technically sound, and do the data support the conclusions?

Reviewer #1: Yes

Reviewer #2: Yes

Reviewer #3: Yes

2. Has the statistical analysis been performed appropriately and rigorously? 

Reviewer #1: Yes

Reviewer #2: Yes

Reviewer #3: I Don't Know

3. Have the authors made all data underlying the findings in their manuscript fully available?

Reviewer #1: No

Reviewer #2: Yes

Reviewer #3: Yes

4. Is the manuscript presented in an intelligible fashion and written in standard English?

Reviewer #1: Yes

Reviewer #2: Yes

Reviewer #3: Yes

5. Review Comments to the Author

Reviewer #1: Dear authors and editor,

Thank you for the opportunity to review this interesting manuscript on the pre-analytical quality of blood cultures. I commend the authors for conducting this extensive project. The study shows that the sex of the patient, the amount of nursing staff, and the type of blood draw are significantly associated with the filling volume of blood culture bottles. However, the volumes did not significantly impact positivity rates. The manuscript is well-written, the statistical analyses seem correct and thorough, and the topic is important. I have some suggestions to improve the manuscript further.

Main comments:

This study may not have included enough patients to draw important conclusions, especially given some very large confidence intervals in the (multivariable) models. Can the authors explain why they included patients for just 58 days in the methods section? Did they carry out a power calculation of some sort? The authors comment on this aspect in the discussion, stating: “the precision of our estimate was most likely not able to detect such small effect sizes due to the sample size in our study.” However, this reviewer feels that this was to be expected beforehand and urges the authors to describe why they still limited the sample size to 58 days of inclusions.

The analyses of the association of age with BC volumes regard age as a continuous variable. In the discussion, it is mentioned that others (Henning et al.) have found age to be an important factor associated with BC volumes, but this study does not. Can you bin the age variable or make it binary around some cut-off (younger/older)? That could still show a significant impact of age.

Minor comments

In lines 60-62: the authors state that the time to effective antibiotic treatment is directly associated with mortality and support that with three references. I would suggest weakening this claim since these studies do not necessarily look at the time of “effective” antibiotic treatment but the time to any antibiotics. On top of that, systematic reviews have not been able to show this effect (e.g., Sterling et al., the impact of timing of antibiotics on outcomes in severe sepsis and septic shock: a systematic review and meta-analysis).

In lines: 165-166: it is stated that the positivity rate in the study was 5.9%, while in line 161 it is stated that the average positivity rate in UHB is 8.4%. Can the authors discuss why their study sample differs so much from the average? Is there some bias that we need to account for here?

In line 167: the authors mention a trend of higher volumes in positive BCs compared with negative BCs. I strongly recommend rephrasing or removing this section since it can be misleading. Firstly, your modeling procedures indicate that this is not the case. Secondly, no p-values are provided, and the term “trend” is vague in this context.

Reviewer #2: This article evaluates the sample volume of blood cultures and factors associated with reaching the recommended volume (>8 mL). In addition, the authors evaluated blood culture positivity and time to positivity.

The results showed that the sample volume of most BC bottles was below the recommended volume. Furthermore, they showed that an increase in the number of nurses was associated with meeting the recommended volume of BC bottles. For some of the content, I recommend that the authors add a discussion based on actual clinical practice.

These data may provide data on the rate of compliance with recommended volumes in blood cultures and may help optimize the collection of blood cultures.

Comment 1, Line 98-99: You should add whether the time from blood sampling to the start of processing was monitored. The time from blood sampling to the start of the BC affects the time to a positive or positive BC. If this time was not monitored, I would recommend adding the impact it had on the results.

Comment 2, Line 181: Add a space between the number and the unit (e.g., 14.9 mL). I recommend that you check out the entire article.

Comment 3, Line 194: Could you please unify the fonts in each table? For example, in table 1, the font for each factor is in italics. On the other hand, in table 2, the font for each factor is normal.

Comment 4, Line 209: In Tables 2 and 3, you should give a heading (patient, case, sample, bottle) for each factor as in Table 1. The headings will help the reader understand the table.

Comment 5, Line 277-280: In clinical practice, do you believe that the number of nurses should be increased to ensure adequate BC volume? In actual clinical practice, there may be many setting where there are not enough nurses. I recommend that you also discuss the issues in clinical practice.

Comment 6, Line 286-288: Based on the study's results, should catheter blood collection be recommended in routine practice? Collection from intravenous catheters may result in higher rates of blood culture contamination. Please add a discussion based on this point.

Comment 7, Line 315: Please be specific about the possible impact of the 9 mL syringe volume on the study results. Could the syringe volume have had either a positive or negative effect on the BC volume?

Reviewer #3: The manuscript titled “Factors Affecting Blood Culture Pre-Analytical Quality – Analysis at a Tertiary Medical Center” by L. Romann et al., deals with the correlation of different types of factors on blood culture quality as a tool for diagnosis of blood infections. The study has some limitations such as the limited number of positive bottles (246/4118), among which 55 are considered contaminants, hence the limited statistical power. As a result, the authors do not have sufficient data to analyze the effects of specific microorganisms. However, the topic of the study is impactful and the current literature is scarce.

The manuscript needs minor revision before publication. My recommendations:

The authors used a precise method to infer the degree of filling of blood culture bottles by weighing. To my knowledge, the Virtuo instrument is capable of flagging bottles that are under volume. Do the authors have any comments on this? 2) There are some terms that need to be better explained in the main text instead of the supplementary topics: Infection status, case and sample. As well as the criteria on which the definition of contaminated blood culture was based. 3) what was the strategy adopted with respect to blood sampling? single or multiple sampling strategy? 4) In table 1, cases section, there are some topics that need to be better explained: "at least one infectious diagnosis", does it refer to which type of infection and when? Does "total sample volume versus case" refer to the single sampling strategy or not? 5) in table 1, bottle section, why did the authors choose to stratify the time to positivity into three categories according to <20 hours, 20-30 hours and >30 hours? 6) in Table 1, bottle section, better specify which type of microorganisms have been considered contaminants and which are Gram positive and Gram negative microorganisms at species level. 7) Among patients admitted to intensive care units (192), does the degree of filling affect the rate of positive blood cultures? 8) In Supplementary Figure 3, why is the maximum value in the x-axis 15?

6. PLOS authors have the option to publish the peer review history of their article (what does this mean?). If published, this will include your full peer review and any attached files.

Reviewer #1: No

Reviewer #2: No

Reviewer #3: No

---

## [Author Response · Author response to Decision Letter 0]

18 Feb 2023

We thank the reviewers and editor for reviewing our paper and providing valuable remarks. We feel that due to your comments the manuscript was improved. The authors have carefully considered the comments and we tried to address every point. Based on the notion of reviewer 1 on the data availability we will upload an anonymized dataset if requested.

The authors welcome further comments if any.

Below we provide the point-by-point responses. The indicated lines in brackets correlate to the lines in the "manuscript" All modifications are visible in the "manuscript with track changes".

Thank you again sincerely,

Lucas Romann

Prof. Adrian Egli

Response to Academic Editor:

1. Please ensure that your manuscript meets PLOS ONE's style requirements, including those for file naming. The PLOS ONE style templates can be found at https://journals.plos.org/plosone/s/file?id=wjVg/PLOSOne_formatting_sample_main_body. pdf and https://journals.plos.org/plosone/s/file?id=ba62/PLOSOne_formatting_sample_title_author s_affiliations.pdf

Response: We made the following adjustments: Level 1, 2 and 3 headings, figure and supporting information titles, double-spacing paragraph format, unifying the tables and adding subheadings for table 2-5 (we did not include all the formatting track changes of the tables since it drastically slowed the Word performance), titles of tables, manuscript title, affiliations and corresponding authors, deleting the acknowledgment section. For further details please have a look at the "Manuscript with track changes".

Response: As requested we gave additional information regarding the ethic statement as follows:

"All methods in this study were carried out in accordance with the Declaration of Helsinki. No study specific consent, but an individual written general informed consent was available. Patients who decline the general consent were excluded. For cases with no positive or no negative statement (unclear status), the need for consent was waived by the ethical committee as there was no ethical concern and this is a paragraph in the Swiss Human Research Act (article 34, “Humanforschungsgesetz”). Therefore, we included all BCs from hospitalized patients ≥ 18 years of age who did not reject the institutional general consent. Samples from children were excluded due to different procedures and age-dependent variability in collected blood volumes.

The project was ethically evaluated and approved by the “Ethikkommission Nordwest- und Zentralschweiz (https://www.eknz.ch/)”, this is the legal body to evaluate projects in our region. The evaluation document had the following number (project-ID: 2020-00451)." (Line 87-97)

3. PLOS requires an ORCID iD for the corresponding author in Editorial Manager on papers submitted after December 6th, 2016. Please ensure that you have an ORCID iD and that it is validated in Editorial Manager. To do this, go to ‘Update my Information’ (in the upper left- hand corner of the main menu), and click on the Fetch/Validate link next to the ORCID field. This will take you to the ORCID site and allow you to create a new iD or authenticate a pre- existing iD in Editorial Manager. Please see the following video for instructions on linking an ORCID iD to your Editorial Manager account: https://www.youtube.com/watch?v=_xcclfuvtxQ

Response: I created an ORCID iD and linked it to my account in the Editorial Manager.

Response: The ethic statement is as of now only written in the Methods section.

Response: We included the Supporting Information at the end of the manuscript and adjusted the titles of the corresponding figures.

Response: Thank you for the kind reminder. We adjusted reference number 13 to the actual article instead of the review which we have cited previously. We adapted the reference library, because reference 15 and 16 are "new media" and separated them with two headings and added the web links.

Response to Reviewer #1:

Thank you for the opportunity to review this interesting manuscript on the pre-analytical quality of blood cultures. I commend the authors for conducting this extensive project. The study shows that the sex of the patient, the amount of nursing staff, and the type of blood draw are significantly associated with the filling volume of blood culture bottles. However, the volumes did not significantly impact positivity rates. The manuscript is well-written, the statistical analyses seem correct and thorough, and the topic is important. I have some suggestions to improve the manuscript further.

Response: Thank you very much for this encouraging assessment.

Main comment. This study may not have included enough patients to draw important conclusions, especially given some very large confidence intervals in the (multivariable) models. Can the authors explain why they included patients for just 58 days in the methods section? Did they carry out a power calculation of some sort? The authors comment on this aspect in the discussion, stating: “the precision of our estimate was most likely not able to detect such small effect sizes due to the sample size in our study.” However, this reviewer feels that this was to be expected beforehand and urges the authors to describe why they still limited the sample size to 58 days of inclusions.

Response: The project started as a master thesis for a medical degree. The period for BC weighing was a month because we expected to have between 3000-4000 positive BCs per year and regarding the primary endpoint, we estimated that a month’s BCs would be enough. Due to the SARS-CoV 2 pandemic the lab was closed for medical students during the lockdown and an extended period of measurement was not possible. We did not carry out a power calculation. We now mention this in the Methods section as follows:

"No prior power calculation was carried out beforehand." (Line 184 and 185)

Main comment. The analyses of the association of age with BC volumes regard age as a continuous variable. In the discussion, it is mentioned that others (Henning et al.) have found age to be an important factor associated with BC volumes, but this study does not. Can you bin the age variable or make it binary around some cut-off (younger/older)? That could still show a significant impact of age.

Response: As in Henning et al., we included age as a continuous variable in the model. We did not dichotomize age into two categories to preserve as much explanatory information and statistical power as possible. There is no established cutoff in the literature to distinguish between younger and older patients and finding such a cutoff based on our data could lead to a spurious significant finding. Please see Altman & Royston, 2006, for a more detailed discussion: https://www.ncbi.nlm.nih.gov/pmc/articles/PMC1458573/. There are several possible reasons why our results may differ from those from Henning et al., including the smaller sample size in our study and the choice of variables in our model.

Minor comment. In lines 60-62: the authors state that the time to effective antibiotic treatment is directly associated with mortality and support that with three references. I would suggest weakening this claim since these studies do not necessarily look at the time of “effective” antibiotic treatment but the time to any antibiotics. On top of that, systematic reviews have not been able to show this effect (e.g., Sterling et al., the impact of timing of antibiotics on outcomes in severe sepsis and septic shock: a systematic review and meta- analysis).

Response: Thank you for the suggested publication. We amended our statement accordingly and added the publication as a reference:

"The time to any antibiotic treatment is associated with mortality [1-4]. For this reason, any delay in the diagnostic work-up may impact the outcome." (Line 55-57)

Minor comment. In lines: 165-166: it is stated that the positivity rate in the study was 5.9%, while in line 161 it is stated that the average positivity rate in UHB is 8.4%. Can the authors discuss why their study sample differs so much from the average? Is there some bias that we need to account for here?

Response: The UHB positivity rate includes all patients (exclusion of patients due to declining the general consent) and multiple samples from patients, which we corrected in our analysis. In addition, we faced the first lockdown during this time, and this had an influence on the hospital occupancy and on the quantity of diagnostic measurements.

We added a statement in the Discussion section as follows:

"The lower positivity rate in the time of BC weighing is due to correction of multiple sampling from patients and exclusion of patients who declined the general consent." (Line 364 and 365)

In line 167: the authors mention a trend of higher volumes in positive BCs compared with negative BCs. I strongly recommend rephrasing or removing this section since it can be misleading. Firstly, your modeling procedures indicate that this is not the case. Secondly, no p-values are provided, and the term “trend” is vague in this context.

Response: We revised the sentence as follows:

"Positive BCs did not show significantly higher volumes compared to negative BCs (7.34 mL, IQR 7.15-7.55 vs. 7.17 mL, IQR 7.13-7.22)." (Line 212-214)

Response to Reviewer #2:

This article evaluates the sample volume of blood cultures and factors associated with reaching the recommended volume (>8 mL). In addition, the authors evaluated blood culture positivity and time to positivity.

The results showed that the sample volume of most BC bottles was below the recommended volume. Furthermore, they showed that an increase in the number of nurses was associated with meeting the recommended volume of BC bottles. For some of the content, I recommend that the authors add a discussion based on actual clinical practice.

These data may provide data on the rate of compliance with recommended volumes in blood cultures and may help optimize the collection of blood cultures.

Response: Thank you very much for your positive evaluation.

Comment 1, Line 98-99: You should add whether the time from blood sampling to the start of processing was monitored. The time from blood sampling to the start of the BC affects the time to a positive or positive BC. If this time was not monitored, I would recommend adding the impact it had on the results.

Response: In the data the time from blood sampling to the start of the processing was not transparent as our hospital does not track the sampling timepoint precisely. We have a loading time available of the BC into the incubator (Virtuo 

system), but unfortunately no exact drawing time – as the nurses do not note this and it is possible that before the transport to the lab a BC waits on a ward for 30min. Yet, from clinical experience we see that these times are rather short. But we do not have a minute resolution on this. We feel that the impact it has on our results is small in our opinion. We added a paragraph as follows:

"The time from blood sampling to the start of the incubation of the BC was not distinguishable in our dataset. However, in our opinion the impact it had on our time to positivity was small because these times are rather short as clinical practice shows us at the UHB." (Line 372-375)

Comment 2, Line 181: Add a space between the number and the unit (e.g., 14.9 mL). I recommend that you check out the entire article.

Response: We revised the manuscript accordingly.

Comment 3, Line 194: Could you please unify the fonts in each table? For example, in table 1, the font for each factor is in italics. On the other hand, in table 2, the font for each factor is normal.

Response: Thank you very much for picking this up. Now, we harmonized the tables.

Comment 4, Line 209: In Tables 2 and 3, you should give a heading (patient, case, sample, bottle) for each factor as in Table 1. The headings will help the reader understand the table.

Response: We added subheadings for table 2-5 and adapted the rows accordingly.

Comment 5, Line 277-280: In clinical practice, do you believe that the number of nurses should be increased to ensure adequate BC volume? In actual clinical practice, there may be many setting where there are not enough nurses. I recommend that you also discuss the issues in clinical practice.

Response: Thank you for your input. In actual clinical practice, increasing the number of nurses would not be justifiable, because the effect is limited and the expenses too costly. On top of that, pre-analytical quality in nursing is important but to quantify this is extremely difficult and our study is potentially one aspect of this. Therefore, we cannot give a clear statement on this. We added the following paragraph to our discussion:

"Therefore, increasing the number of nurses in clinical practice would not be justifiable, because the effect is limited and the expenses too costly. Moreover, it is not the only reason for the lower filling volumes." (Line 325-327)

Comment 6, Line 286-288: Based on the study's results, should catheter blood collection be recommended in routine practice? Collection from intravenous catheters may result in higher rates of blood culture contamination. Please add a discussion based on this point.

Response: We added a point in the discussion as follows:

"Yet, they are more costly and might require a specialist placing them, which reflects just the real-world experience. Nonetheless, contaminations of intravenous catheters have been successfully reduced in the last years with different methods [20][21]. To evaluate the cost- effectiveness further studies are needed." (Line 333-336)

Comment 7, Line 315: Please be specific about the possible impact of the 9 mL syringe volume on the study results. Could the syringe volume have had either a positive or negative effect on the BC volume?

Response: Thank you for the comment. Due to this limit, we cannot calculate if there is an effect of the syringe volume on the filling volume of the BC. We amended the paragraph in the discussion as follows:

"Therefore, they are most likely part of the reason for the low filling volume in more than half of the BC bottles." (Line 360 and 361)

Response to Reviewer #3:

The manuscript titled “Factors Affecting Blood Culture Pre-Analytical Quality – Analysis at a Tertiary Medical Center” by L. Romann et al., deals with the correlation of different types of factors on blood culture quality as a tool for diagnosis of blood infections. The study has some limitations such as the limited number of positive bottles (246/4118), among which 55 are considered contaminants, hence the limited statistical power. As a result, the authors do not have sufficient data to analyze the effects of specific microorganisms. However, the topic of the study is impactful and the current literature is scarce.

The manuscript needs minor revision before publication.

Response: Thank you very much for your positive feedback.

My recommendations: The authors used a precise method to infer the degree of filling of blood culture bottles by weighing. To my knowledge, the Virtuo instrument is capable of flagging bottles that are under volume. Do the authors have any comments on this?

Response: Yes, this is a newly introduced feature in the Virtuo instrument. It requires a very precise placement of the stickers and is only categorical (above or below a fixed volume). Independent of this study, we have investigated this feature, but the stickers are very often wrongly placed. Also, it will only provide a categoric cut-off and not a continues variable. For this publication and master thesis, we have used therefore individual weighting of the blood cultures.

2) There are some terms that need to be better explained in the main text instead of the supplementary topics: Infection status, case and sample. As well as the criteria on which the definition of contaminated blood culture was based.

Response: We included the definitions and contamination criteria in the manuscript. (Line 141-176)

3) what was the strategy adopted with respect to blood sampling? single or multiple sampling strategy?

Response: The standard procedure at the UHB is the single sampling strategy. We revised the sentence as follows:

"Internal hospital standard operating procedures specify that nursing staff should fill two Monovette® tubes (Sarstedt, Nuembrecht, Germany) with 8-10 mL of blood by single sampling strategy taken under aseptic conditions without air supply and transfer the blood from each tube to an aerobic and an anaerobic BC." (Line 100-103)

4) In table 1, cases section, there are some topics that need to be better explained: "at least one infectious diagnosis", does it refer to which type of infection and when? Does "total sample volume versus case" refer to the single sampling strategy or not?

Response: "At least one infectious diagnosis" does refer to if there was an infection during the whole hospital stay of a patient according to the ICD 10 diagnosis with which the patient has been labeled but not which type of infection. "Total sample volume" includes blood from an aerobe and an anaerobe bottle, while "total case volume" includes the blood collected for BCs during the whole length of hospital stay. It does not refer to the single or multiple sampling strategy. We added the definitions in the corresponding paragraph as follows:

"Total sample volume: Includes blood from an aerobe and an anaerobe BC bottle.

Total case volume: Includes the blood of a patient collected for BCs during the whole length of a hospital stay." (Lines 144-146)

"At least one infectious diagnosis: There has been at least one infection during the whole hospital stay of a patient according to the ICD 10 diagnosis with which the patient was labeled. It does not differ regarding the type of infection." (Line 151-153)

5) in table 1, bottle section, why did the authors choose to stratify the time to positivity into three categories according to <20 hours, 20-30 hours and >30 hours?

Response: We choose the categories according to the distribution of our time to positivity data. Because of the rather few data we had we decided to implement the time to positivity as a categorical variable.

6) in Table 1, bottle section, better specify which type of microorganisms have been considered contaminants and which are Gram positive and Gram negative microorganisms at species level.

Response: We included the microorganisms which we have considered contaminants in the paragraph about contamination (Line 173-176). Additionally, we created a list with all microorganisms which we will upload as a supplementary file. (S1 Table)

7) Among patients admitted to intensive care units (192), does the degree of filling affect the rate of positive blood cultures?

Response: This would be a subgroup analysis which after careful consideration with our statistician, we should not include in our manuscript due to the low sample number and because we have not designed the study to pick up specific difference in the ICU. But we made a specific analysis to inform the reviewer and share the result just with him/her (table is attached in the "Response to Reviewers").

8) In Supplementary Figure 3, why is the maximum value in the x-axis 15?

Response: Our model was developed using data that included almost no bottles filled with more than 10 ml. In Supplementary figure 3 (S3 Fig), we used this model to project what the effect for volumes above 10 ml could yield (with wider bands of uncertainty).

---

## [Editor Report · Decision Letter 1]

27 Feb 2023

Factors impacting the pre-analytical quality of blood cultures - analysis at a tertiary medical center.

PONE-D-23-00409R1

Dear Dr. Romann,

We’re pleased to inform you that your manuscript has been judged scientifically suitable for publication and will be formally accepted for publication once it meets all outstanding technical requirements.

Kind regards,

Vittorio Sambri, M.D., Ph.D.

Academic Editor

PLOS ONE

Additional Editor Comments: all of the points  raised and the suggestions made by the three reviewers have been substantially answered and considered.
---

## [Editor Report · Acceptance letter]

6 Mar 2023

PONE-D-23-00409R1 

Factors impacting the pre-analytical quality of blood cultures - analysis at a tertiary medical center. 

Dear Dr. Romann:

I'm pleased to inform you that your manuscript has been deemed suitable for publication in PLOS ONE. Congratulations! Your manuscript is now with our production department. 

Kind regards, 

on behalf of

Professor Vittorio Sambri 

Academic Editor

PLOS ONE